# Analysis and Evaluation of Dental Caries in a Mexican Population: A Descriptive Transversal Study

**DOI:** 10.3390/ijerph20053873

**Published:** 2023-02-22

**Authors:** Alejandro Moreno-Barrera, Pedro Morales-Ruiz, David Ribas Pérez, Javier Flores-Fraile, Antonio Castaño-Seiquer

**Affiliations:** 1Department of Surgery, University of Salamanca, 37008 Salamanca, Spain; 2Department of Stomatology, University of Seville, 41004 Seville, Spain

**Keywords:** caries prevalence, epidemiology, dental public health

## Abstract

Oral diseases are an important public health problem owing to their high prevalence and strong impact on people, particularly in disadvantaged populations. There is a strong relationship between the socioeconomic situation and the prevalence and severity of these diseases. Mexico is among the countries with a higher frequency range in oral diseases, highlighting dental caries, which affect more than 90% of the Mexican population. Materials and method: A cross-sectional, descriptive, and observational study was carried out in 552 individuals who underwent a complete cariogenic clinical examination in different populations of the state of Yucatan. All individuals were evaluated after providing informed consent and with the consent of their legal guardians for those under legal age. We used the caries measurement methods described by the World Health Organization (WHO). Prevalence of caries, DMFT, and dft indexes were measured. Other aspects were also studied, such as oral habits and the use of public or private dental services. Results: The prevalence of caries in permanent dentition was 84%. Moreover, it was found to be statistically related to the following variables: place of residence, socioeconomic level, gender, and level of education (*p* < 0.05). For primary teeth, the prevalence was 64% and there was no statistical relation with any of the variables studied (*p* > 0.05). Regarding the other aspects studied, more than 50% of the sample used private dental services. Conclusions: There is a high need for dental treatment in the population studied. It is necessary to develop prevention and treatment strategies considering the particularities of each population, driving collaborative projects to promote better oral health conditions in disadvantaged populations.

## 1. Introduction

Oral diseases constitute a significant public health problem because of their high prevalence and strong impact on people and society in terms of pain, social, and functional disability [1]. Currently, nine out of ten people in the world are at risk of suffering from an oral disease [2,3,4,5].

Mexico is among the countries with a high frequency range in oral diseases. The prevalence of caries affects more than 90% of the Mexican population [6]. According to the Universal Catalogue of Health Services (CAUSES), the Mexican state offers medical coverage that also includes dental specialties [7].

If we focus on the Yucatan region (which belongs to the south-eastern area of Mexico), according to data from the General Direction of Epidemiology of the Ministry of Health of the Government of Mexico, in 2019, the region had rates of oral diseases comparable to the rest of the country and markedly inferior to other North American (United States of America) or European (United Kingdom or Sweden) countries [8].

Thus, we find an exaggerated high proportion of children with ECC receiving health services (31.8 vs. 6% in the USA) and a caries index (DMFT) at 12 years of age of 2.6 vs. 1.2 in the USA or 0.8 in Sweden. Although in adults aged 35–44 years, the data are similar in terms of DMFT, the percentage of fillings (the so-called restoration index) is markedly higher in the countries mentioned above compared with the Yucatan region (20% compared with 52% in the United Kingdom or 63% in the USA). Even greater is the difference in terms of edentulism or lack of functional occlusion in adults aged 65–74 years (55.8% of non-functional occlusion in the southeast region of Mexico compared with 38.2% in the USA) [9,10,11].

Worldwide, the incidence of oral diseases, particularly in disadvantaged populations, remains high [12,13]. Among the main ones, we highlight decayed teeth as the most prevalent, followed by periodontal conditions, malocclusions, and oral trauma, which affect the quality of life of those who suffer from it [2].

Dental caries is defined as a multifactorial chronic disease, which develops under the following conditions; a susceptible host; a cariogenic oral flora; and an appropriate substrate that must be present for a specified period of time and that, in turn, will be influenced by the community, family, and individual predisposition [14,15,16,17].

Caries experience is the number of teeth/surfaces that have caries lesions (at a specified threshold), restorations, and/or are missing owing to caries, accumulated by an individual up to a designated point in time. Though new models or indexes are being explored internationally, the majority of studies measure the caries experience by means of DMFT/S (dft/s) at varying detection levels [18].

Peres and Cos (2009) stated that there is a very strong and persistent relationship between socioeconomic status and the prevalence and severity of oral diseases [3,12]. This is usually linked to a cariogenic diet and poor oral hygiene, as well as the consumption of tobacco and alcohol and low accessibility to oral health services. In addition to other factors such as dental malposition, parental education and associated systemic diseases usually coexist with a lack of oral hygiene [12,19,20,21].

Concerning caries, prevention through proper oral hygiene, a non-cariogenic diet, and topical fluoride is the most effective method to decrease its development. Early detection would also avoid severe complications such as advanced caries, pulpitis, endodontic treatments, and loss of teeth [22]. Caries prevention has traditionally meant inhibition of caries initiation, otherwise called primary prevention. Primary, together with secondary and tertiary prevention, comprise non-operative and operative treatments for caries management [18].

The main objective of this research was to analyse the prevalence and index of dental caries in primary and permanent dentition defined by type of population, rural or urban, among populations of the state of Yucatan, Mexico.

## 2. Materials and Methods

### 2.1. Study Type and Settings

An observational, cross-sectional, and descriptive study carried out as part of the “Yucatán International Cooperation Project” was developed in Temax, Hunucmá, Umán, and Mérida.

The study sample consisted of 552 individuals between 5 and 64 years old who requested dental care. All participants signed an informed consent form and filled out an individual survey on oral health, oral hygiene habits, and quality of life. The information from underage patients was collected by their parents or legal guardians once the consent was signed. In addition, each individual underwent a complete clinical dental examination focused on cariogenic pathology.

The World Health Organization (WHO) criteria for dental caries and care needs related to the condition of the teeth were applied [23]. All of the participants in the study were examined in natural light and a no. 5 flat mirror was used. The participants brushed their teeth before the examination and the teeth were not dried prior to the inspection.

All patients had the same clinical examiner (A.M.), with the same methodology used on all of them them to avoid bias. It was decided to carry out the scanning for data collection through the work from a single examiner (a dentist with lots of experience on caries assessment). With the aim to measure the consistency of the observations, the examiner was subjected to a so-called intra-observer calibration, obtaining the ratio of agreement with a Kappa test (0.85).

### 2.2. Study Variables

The variables analysed were age, sex, place of residence (urban or rural area), socioeconomic level (low rural, low urban, or urban environment), and highest level of studies achieved.

The variables obtained from the questionnaire on oral health attitudes and habits and the use of dental health services were also studied. For the clinical variables of caries indexes, the prevalence of caries and the DMFT index for permanent dentition and dft for primary dentition were studied according to WHO criteria of caries [23].

### 2.3. Statistical Analysis

Statistical analysis was performed using STATA V15 (College Station, TX, USA). Continuous variables were summarized through means and standard deviations (SDs). The categorical variables are presented through the frequency distribution and the simple and cumulative frequencies are reported in percentages. 

Associations between prevalence and cavity rates were studied with age, sex, area of origin, socioeconomic level, and education. ANOVA was used for continuous variables and the chi-square test was performed for categorical variables. The critical value to identify statistically significant differences was *p* < 0.05.

## 3. Results

### 3.1. Sociodemographic Data

The mean age of the population was 28.8 ± 16.2. Four age groups were categorized: children 6–12 years, adolescents 12–19 years, young adults 20–34 years, and older adults 35–64 years. Among the age groups chosen for the study, the so-called “older adults”, made up of subjects between 35 and 64 years of age, represented the largest group, with 37.32%. In terms of gender, women accounted for 60.51% of the sample, while men accounted for 39.49%. 

Concerning other socio-demogaphic data, fifty-six percent of the population studied was rural, with a low socioeconomic level. Twelve percent of the sample reported having no education. Of the 92% who had completed studies, 83% had completed primary, secondary, or high school and only 17% had reached university level. All these data are summarised in Table 1.

### 3.2. Oral Health Attitudes and Practices

The following are some of the most significant results from the oral health attitudes and practices survey of the study participants.

More than half of the surveyed population is very concerned about their oral health (58%), while 7% of them acknowledge having little concern for their oral health (Table 2).

Fifty-four percent of individuals reported brushing three times a day, 34.8% twice a day, and 7% only once a day. In the group of women, 55.9% reported brushing three times a day and this percentage was lower in the group of men (38.5), (*p* < 0.001) (Table 3).

Furthermore, 90.45% of the study population used a manual toothbrush. Only three subjects used an electric toothbrush. Sixty-four percent of the population used a toothpaste as the main complementary product for toothbrushing. Almost one in five subjects used mouthwash, while dental floss was only used by 12% of the population (Table 4).

### 3.3. Use of Dental Services

With regard to the frequency of dental check-ups, it can be seen that the most frequently answered response by the population to the question regarding when they should visit their dentist was “when they have a problem” (Table 5).

Moreover, 8.15% of the population acknowledged that they had never visited a dentist, while 20.29% had done so less than six months ago. The rest of the population under study had visited their dentist more than one year ago (Table 6).

More than half of the population used private services, while 35.59% used the public dental services made available by the state. Ten percent stated that they were not aware of the difference between the two types of care (Table 7).

### 3.4. Dentition Status

For primary dentition, the prevalence of caries was 64%, with a dft value of 2.8 ± 3.19. More specifically, for children aged 5 and 6 years old, the prevalence obtained was 55%, obtaining a dft value = 2.45 ± (3.21). Analyzing the relationship between the dft index and the socio-demographic variables, it is observed that there is no statistical significance (*p* > 0.05 in all cases) (Table 8 and Table 9).

Regarding permanent dentition, 94% of the population (*n* = 520) had at least one permanent tooth in the mouth. The prevalence of caries was 84%, with a DMFT value of 6.3 ± 0.24. The prevalence of caries in individuals aged 12 years was also calculated, which was 54%, with a DMFT value of 1.1 (±1.11). Regarding the relationship between the DMFT index and socio-demographic variables, it is observed that there is a statistically significant relationship between DMFT and age, type of residence, socioeconomic level, and educational level (*p* < 0.05) (Table 10 and Table 11).

## 4. Discussion

According to the WHO, dental caries is the most prevalent disease in the world, affecting more than 80% of the world’s population, in addition to being considered the most prevalent pathology in the child population [24,25]. 

For the selection of our simple, and despite having studied a relatively large number of patients, we took a number of patients who came to the Yucatan International Cooperation Project requiring dental care. This could be a limitation of our study, as they were people with a perceived need for treatment and thus would not be representative of the whole population, which is a limitation to be taken into account when interpreting the results. Nevertheless, the authors believe that it serves to show a snapshot of the oral health of the Yucatecan population.

### 4.1. Primary Dentition

The prevalence of carious lesions in primary dentition in the Yucatecan population studied was 64%, a figure similar to that in the work carried out by the Mexican group of Montero and Cols. [26]. According to the results of Martínez-Pérez and Cols. and Serrano-Piña and Cols., 5 out of 10 children present caries in primary dentition, while for Villalobos and Cols. or García Pérez and Cols., up to 9 out of 10 children have dental caries [27,28,29,30].

In the present investigation, no statistically significant differences (*p* > 0.05) were found, but a notable association between a higher prevalence of decayed teeth and the rural environment was found, matching with previous studies [30]. Regarding gender, a higher prevalence of caries was found in the female sex and, with respect to the socioeconomic level, 81% of individuals who presented decayed primary teeth belonged to the “low rural” level; in this case, no statistically significant differences were found with the prevalence of caries, like those confirmed by Frencken et al. in their study [31].

The total value of dft for the studied population was 2.8 ± 3.19. The carious component was 2.69 ± 3.08 and the filled component was 0.11 ± 0.11, which shows a high need for treatment, finding more than two untreated caries in each individual. The value of our results with respect to dft in rural areas is similar to those of Medina-Solís and Cols. They obtained a dft of 2.86 in a sample with children from 6 to 12 years old in a non-urban area in Campeche, a state adjacent to Yucatán. In relation to the urban areas, we obtained a value of 2.7 for dft, compared with the value of 2.4 obtained for the dft index in the above-mentioned study. It is important to note that, if teeth lost as a result of caries had been considered, the dft value could have increased and could resemble the results of Romo and Cols. or Villalobos and Cols. [29,32,33].

Although no statistically significant differences were found with the socio-demographic variables, the authors observed that the value of the dft index was higher in men (3.1) than in women (2.3); that the “low rural” socioeconomic group obtained better dft index values than the “low urban”; and, in terms of schooling, individuals without schooling obtained higher levels of dft than those who had studied primary school, at 3.6 and 2.3, respectively.

Following the WHO instructions, 5- and 6-year old children were specifically studied. Although the sample volume is low (n = 22), given that the data collection was carried out in a random sample, the prevalence was 55%, presenting a dft of 2.45 ± 3.21. The results of the present research in terms of prevalence are similar to those of the National Dental Caries and Fluorosis Surveys in Mexico [34]. In the case of dft, the difference is more evident, as it was greater in our study than in the results of the survey (1.5 vs. 2.45). Regarding the decayed component, the value obtained in our research was 2.36 ± 3.23, while the that of the national survey was 1.3. This could be justified because the target population of the project was the one with the least economic resources. The sealed component was 0.09 ± 0.29, highlighting the existing treatment needs in this sector.

### 4.2. Permanent Dentition

On the other hand, the prevalence of caries in permanent dentition was 84%, a result similar to that of other studies with the same methodology carried out in other regions of Mexico, Romo and Cols. in Nezahualcóyotl, Aamodt and Cols. in Chiapas, and Islas-Granillo and Cols. in Hidalgo [30,32,35,36].

The results led to a greater presence of decayed teeth in urban populations compared with rural ones (*p* = 0.027), a fact that coincides with the results published by Ortega-Maldonado et al. [37]. Regarding the socioeconomic level, there was a significant association between this variable (*p* = 0.031) and the prevalence of caries; the lower income level coincides with a greater presence of this pathology, which agrees with other studies published in Mexico by Villalobos-Rodelo and Cols., in addition to the one published by Vega-Lizama and Cols. [12,38,39,40].

In relation to schooling, a statistically significant relationship was found with the prevalence in permanent dentition (*p* = 0.001), as well as with gender (*p* < 0.05), as it was more prevalent in the female sex; these results coincide with other international studies [33].

The value of the DMFT index obtained in our population was 6.3 ± 0.24. Regarding this, the decayed component was 4.1 ± 3.91, the absent one was 1.3 ± 2.78, and the obturated one was 0.9 ± 2.16. According to our study, only one in six decayed teeth are filled in the population of our study, showing once again that the need for dental care in our sample is high. Other results obtained by Mexican researchers such as Aamot and Cols. only corroborate the high demand for care that exists in this population [35,39].

The DMFT value was higher in women, a fact that coincides with other studies such as that of Romo and Cols. In addition, statistically significant differences (*p* < 0.05) were found with the age variable, with a clear tendency for the DMFT value to increase over time, coinciding with other results published in international literature [32,37]. Regarding the variable place of residence, the urban population obtained higher DMFT levels than the rural population (*p* < 0.05). Regarding the socioeconomic level, the population of the low urban group obtained the highest DMFT value (*p* < 0.05). Both statistically significant associations may be due to the greater access, by the urban population, to products with large amounts of refined sugars and the high intake of carbonated beverages that occurs in this sector [38,40].

Following the WHO instructions concerning age analysis, 12 year olds were also specifically studied and, although the sample size (n = 26) was low because of the random selection of the population, the prevalence of caries was 46%, with a DMFT value of 1.1 ± 1.11. The decayed component was 1.0 ± 1.75, that of missing teeth was 0.38 ± 0.19, and that of filled teeth was 0.03 ± 0.19. The data obtained in the last National Dental Caries and Fluorosis Survey in Mexico for the 11–12-year-old group showed a 47% prevalence of caries in permanent dentition and a DMFT of 1.5, coinciding almost exactly with the results obtained in the present study [34].

### 4.3. Oral Health Habits and Use of Dental Health Services

Oral health is a determinant of quality of life and the acquisition of preventive habits such as toothbrushing can reduce a large number of oral problems. This adoption of preventive habits clearly increases the likelihood of being in optimal health and has been shown to be significantly influenced by socioeconomic and demographic factors [4,5].

With this initial premise, basic issues such as the frequency of tooth-brushing three times a day and the use of fluoride toothpaste are widespread among the population. However, there is still a sector of the population (the so-called fourth world) that, for different socio-economic and/or cultural reasons, has markedly lower rates of oral hygiene habits than the rest of the population [4,5]. 

In our study, we have seen a clear relationship between the level of education and the caries index, with a statistically significant difference between both aspects (*p* < 0.001), which is in agreement with similar studies in other areas of Latin America or the rest of the world [41,42,43].

It is, therefore, socio-cultural issues that mark the acquisition of health habits in general and oral health in particular. In our sample, around 90% of the population brushed their teeth more than twice a day. This percentage is very similar to that of populations with a higher socioeconomic level, such as the Swedish [43] or Spanish [41] population. It seems that these habits are widely acquired by the Yucatecan population.

The use of topical fluorides in the form of toothpaste or mouthwash has been shown to be a major advance in caries control and their widespread use has lowered caries rates worldwide. The percentage of people using fluoride products in our sample was around 80%, somewhat lower than in the countries mentioned above [41,42,43].

With regard to the use of dental health services, the study population reported using mostly private clinics, having visited a dentist in the last two years at a rate of around 60%. The use of these health services is markedly different in other countries. In the USA, almost the entire population uses private dental services, while in European countries, this percentage varies according to the portfolio of services offered by the different countries [41,42,43].

## 5. Conclusions

The present investigation reflects a high need for treatment in the served area of Yucatan, finding more than two untreated caries per individual, with a significantly higher prevalence of decayed teeth in rural areas, among those with low-income levels, and in women. It is crucial to generate a prevention and treatment strategy considering the particularities of each population, improving collaborative projects to promote better oral health conditions not only in the Mexican population, but also in the international arena.

## Figures and Tables

**Table 1 ijerph-20-03873-t001:** Distribution per socio-demographic data.

		Frequency	Percentage
Gender	Male	218	39.49%
	Female	334	60.51%
Age Group	Children (6–12)	111	20.11%
	Adolescents (13–19)	76	13.78%
	Young adults (20–34)	159	28.80%
	Older adults (35–64)	206	37.32%
Educational Level	No studies	69	12.50%
	Primary	116	21.01%
	Secondary	109	19.75%
	High School	162	29.35%
	University	96	17.39%

**Table 2 ijerph-20-03873-t002:** Concerns about oral health.

How Much Do You Care about Your Oral Health?	Frequency	Percent.
A lot	319	57.79%
Regular	104	18.84%
A little	97	17.57%

**Table 3 ijerph-20-03873-t003:** Mechanical plaque control (frequency of toothbrushing).

How Many Times a Day Do You Brush Your Teeth?	Freq.	Percent.
3 times a day	269	49.00%
Twice a day	204	37.16%
Once a day	54	9.84%
Never	22	4.01%

**Table 4 ijerph-20-03873-t004:** Use of oral hygiene products.

Please Specify If You Use Any of the Following Products in Your Dental Care	Freq.	Percent.
None	1	0.18%
Toothpaste	352	63.59%
Mouthrinse	98	17.75%
Dental floss	67	12.14%
Interproximal brush	17	3.08%
Mouth irrigator	2	0.36%

**Table 5 ijerph-20-03873-t005:** Frequency of dental check-ups.

How Often Do You Think It Is Necessary to Visit the Dentist?	Freq.	Percent.
When I have a problem	154	27.90%
Every 2 years	108	19.57%
Every year	112	20.29%
Every 6 months	83	15.04%
Monthly	40	7.25%
Don’t know	54	9.78%

**Table 6 ijerph-20-03873-t006:** Last visit to the dentist.

When Was the Last Time You Visited the Dentist?	Freq.	Percent.
Less than 6 months ago	112	20.29%
1 or 2 years ago	213	38.59%
More than 2 years ago	178	32.25%
Never	45	8.15%
Don´t Know	2	0.36%

**Table 7 ijerph-20-03873-t007:** Type of dental office.

What Type of Clinic Did You Attend?	Freq.	Percent.
Private	276	54.87%
Public	179	35.59%
Don’t know the difference	48	9.54%

**Table 8 ijerph-20-03873-t008:** Relationship of socio-demographic variables with the dft index. Significance for *p*-value < 0.05.

	Category	dft	SD	*p*-Value
Gender	Male	3.1	3.60	0.280
	Female	2.3	2.52	
Population	Urban	2.7	4.05	0.864
	Rural	2.8	3.00	
Socioeconomic level	High	1.6	2.61	0.467
	Medium	3.6	4.82	
	Down	2.8	3.00	
Educational level	No studies	3.6	3.47	0.067
	Primary	2.3	2.92	

**Table 9 ijerph-20-03873-t009:** Average number of decayed and filled teeth in primary dentition (dft).

Children with Primary Dentition	Decayed	Filled	dft	CI (95%) dft
81	2.69 ± 3.08	0.11 ± 0.11	2.8 ± 3.19	2.12–3.50

**Table 10 ijerph-20-03873-t010:** Average number of decayed, missing, and filled teeth in permanent dentition (DMFT).

Participants	Decayed	Missing	Filled	DMFT	CI (95%) DMFT
522	4.1 ± 3.91	1.3 ± 2.78	0.9 ± 2.16	6.3 ± 0.24	5.81–6.76

**Table 11 ijerph-20-03873-t011:** Relationship of socio-demographic variables with the DMFT index. Significance for *p*-value < 0.05.

	Category	DMFT	SD	*p*-Value
Gender	Male	5.52	5.40	0.130
	Female	6.77	5.52	
Age	(6–12)	0.96	1.47	0.001
	(13–19)	3.94	3.75	
	(20–34)	6.26	6.26	
	(35–64)	9.39	5.61	
Population	Urban	7.56	5.62	0.001
	Rural	5.23	5.19	
Socioeconomic Level	High	6.48	5.41	0.001
	Medium	8.36	5.70	
	Down	5.23	5.19	
Educational Level	No Studies	8.62	7.43	0.001
	Primary	4.57	5.43	
	Secondary	6.12	5.19	
	High School	6.04	4.92	
	University	7.95	5.21	

## Data Availability

The data presented in this study are available on request from the corresponding author.

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
