# Peer review of "Analysis and Evaluation of Dental Caries in a Mexican Population: A Descriptive Transversal Study"

_ijerph, 2023, doi:10.3390/ijerph20053873_

Round 1

Reviewer 1 Report

Dear authors,

Congratulations to your impressing study! It was interesting to read your manuscript on dental caries in the area of Yucatan in Mexico. Your manuscript shows the burden of dental caries being present in this area and highlights the need for caries management strategies to reduce the number of affected persons. I recommend considering the following suggestions during revision:

1.       Abstract:

-          I kindly recommend adding the caries measurement method (WHO criteria; dmft/DMFT) to the abstract.

-          Could you please add p-values to our results?

2.       Introduction:

-          Line 38ff (definition of cavities and the use of this term throughout the manuscript): Do you mean dental caries or cavitated carious lesions by "cavities"? I recommend using the terminology published by ORCA and Cariology Research Group of IADR (Machiulskiene et al., Caries Res, 2019; DOI: 10.1159/000503309) when referring to dental caries.

3.       Methods:

-          Line 61: You mention that information about oral health, oral hygiene, and quality of life were gathered. Could you add some of these results to the manuscript?

-          Line 66: "(ref.)" I think there is a reference missing.

-          How many examiners performed the dental examination? How experienced were they in caries assessment? Were the examiners calibrated with regard to caries assessment?

-          Line 66-67: You describe how the dental examination of schoolchildren was performed – how was is done in the other age group? Were the teeth brushed before examination?

-          Is there any information about initial carious lesions available from you study?

4.       Results:

-          Could you please translate the Spanish terms in the tables into English? Thank you!

-          Table 1: The percentage is 120.11 in total. I assume there is a calculation error in the percentage of participating adolescents (13.78 % instead of 33.88 %). Could you please check this?

-          Table 5: Is there any information about missing primary teeth for the dmft index? I recommend using the term "primary dentition" instead of "temporal dentition".

-          Table 6, left column: Are there only "adults" included or participants of all age groups with permanent teeth? Could you please use the terms "decayed" and "missing" when referring to the dmft/DMFT index (also in Table 5)?

5.       Discussion:

-          The abbreviation "cod" is used several times. Is it the Spanish term for dmft/DMFT?

Kind regards!

Reviewer 2 Report

This manuscript is entitled “Analysis and evaluation of dental caries in a Mexican population.  A descriptive transversal study”.  The research question being asked is what is the prevalence of dental caries in the state population of Yucatan, Mexico.

The study uses a cross-sectional design of 552 participants who received dental examinations.  Unfortunately, there is no information about sampling and how this sample may be reflective of the overall population of the state.  In fact, if the wording is read to be as accurate, participants requesting dental care as part of the Yucatan International Cooperation Project was the form of sampling.  If this is the case, the results are more likely to over-represent the needs of the community due to perceived treatment need.  This should be clarified based on further discussion.

Despite the collection of a significant amount of information, only examination data related to caries experience is discussion.  However, it is unclear from the methodology discussed if examinations were standardised.  This could have significant influence over caries detection and needs to be discussed in the manuscript.

In relation to the results, the presentation is variable.  The age groups used are highly variable with no consistency in these ranges.  This was likely based on the sample size and to ensure equal proportions, but the age ranges limit useful interpretation of the results.  There are significant differences and diversity in some of the populations, for example, risk factors for oral conditions between adults of age 35 and 64 would be incredibly different.  This should either be justified, but given the results are primarily descriptive, the research team should consider reanalysis.  However, this presents just an example of inconsistencies throughout the results.  For example, further details is need as to how rural vs urban populations were identified in the sample to improve reproducibility of the results.

Despite this, overall, the conclusions are a reasonable reflection of the results.  However, it does not necessarily answer the research question identified.  The manuscript could be improved by language editing.  Although the grammar is accurate, the meaning is sometimes obscured by the words chosen.  Likewise, while studies are not expected to be perfect, because conducting research is always a challenge, the deficiencies of the study should be discussed to ensure these have been acknowledged as part of the research process.

Please see below some more specific examples to consider: 

An example of the language issues is in the Abstract, line 11 “Mexico is among the countries with the a higher frequency range in oral diseases, highlighting cavities, which affect more than 90% of the Mexican population”

For the introduction, is there available population data for Mexico?  It would be beneficial for this to be discussed further to provide context to the study and result.  In addition, the research team should consider why the population in Yucatan is of interest, other than convenience of available data?

Missing reference line 66

Table 1: Title of first column should be in English

Table 4: Scolarity is not a word.  This needs to be replaced

Table 5: Title refers to temporal dentition.  This should either be changed to primary or deciduous dentition which are accepted terms.

Table 7. Populations is spelt wrong.  See comment about about Scolarity.

Line 127.  Discusses cod.   This needs to be clarified.  Likewise cpod in line 133.

No concerns with disclosure statements, but please review formatting of references

Round 2

Reviewer 2 Report

Thank you for updating your manuscript.  Please ensure that the manuscript is revised for language errors.  There are still sections where it is written in Spanish.  Also ensure all abbreviations are introduced at their first time in the manuscript.

Author Response

Dear Sirs, 

Thank your for your comments. We have revised the manuscript changing the sections in Spanish.

Kind regards